# Increased Levels of ICOS and ICOSL Are Associated to Pulmonary Arterial Hypertension in Patients Affected by Connective Tissue Diseases

**DOI:** 10.3390/diagnostics12030704

**Published:** 2022-03-13

**Authors:** Mattia Bellan, Francesco Murano, Federico Ceruti, Cristina Piccinino, Stelvio Tonello, Rosalba Minisini, Ailia Giubertoni, Daniele Sola, Roberta Pedrazzoli, Veronica Maglione, Giulia Francesca Manfredi, Antonio Acquaviva, Roberto Piffero, Giuseppe Patti, Mario Pirisi, Pier Paolo Sainaghi

**Affiliations:** 1Department of Translational Medicine (DiMeT), Università del Piemonte Orientale (UPO), 28100 Novara, Italy; 20035126@studenti.uniupo.it (F.M.); 10036665@studenti.uniupo.it (F.C.); stelvio.tonello@med.uniupo.it (S.T.); rosalba.minisini@med.uniupo.it (R.M.); ailia.giubertoni@maggioreosp.novara.it (A.G.); 20018219@studenti.uniupo.it (V.M.); 20030060@studenti.uniupo.it (G.F.M.); 20030227@studenti.uniupo.it (A.A.); 20009551@studenti.uniupo.it (R.P.); giuseppe.patti@med.uniupo.it (G.P.); mario.pirisi@med.uniupo.it (M.P.); pierpaolo.sainaghi@med.uniupo.it (P.P.S.); 2Rheumatology Unit, Department of Internal Medicine, AOU Maggiore della Carità, 28100 Novara, Italy; daniele.sola@maggioreosp.novara.it (D.S.); roberta.pedrazzoli@maggioreosp.novara.it (R.P.); 3Center for Translational Research on Autoimmune and Allergic Disease (CAAD), Università del Piemonte Orientale (UPO), 28100 Novara, Italy; 4Division of Cardiology, AOU Maggiore della Carità, 28100 Novara, Italy; cristina.piccinino@maggioreosp.novara.it

**Keywords:** ICOS, ICOSL, pulmonary arterial hypertension, systemic sclerosis, connective tissue diseases

## Abstract

Background: Pulmonary hypertension (PH) is a life-threatening complication of connective tissue diseases (CTD); in this study, we aimed at investigating the potential role of inducible co-stimulator (ICOS) and its ligand (ICOS-L) as biomarkers of PH in CTD. Materials and Methods: We recruited 109 patients: 84 CTD patients, 13 patients with CTD complicated by pulmonary arterial hypertension (PAH), and 12 subjects with PAH alone. All recruited patients underwent a complete clinical and instrumental assessment along with quantitative measurement of serum ICOS and ICOS-L. Results: Independently of the underlying cause, patients with PAH were older and had a lower glomerular filtration rate. Interestingly, patients with both CTD-related and CTD-unrelated PAH had higher ICOS and ICOS-L serum concentrations than CTD patients (0.0001 for both). When compared to CTD patients, those affected by CTD-PAH showed higher ICOS (440 (240–600) vs. 170 (105–275) pg/mL, *p* = 0.0001) and ICOS-L serum concentrations (6000 (4300–7000) vs. 2450 (1500–4100) pg/mL; *p* = 0.0001). In a logistic regression, ICOS and ICOS-L were associated with a diagnosis of PAH, independently from age, gender, and renal function. The corresponding receiver operating characteristic (ROC) curves demonstrated a good diagnostic performance for both ICOS and ICOS-L. Conclusions: ICOS and ICOS-L are increased in patients with PAH, irrespectively from the underlying cause, and represent promising candidate biomarkers for the diagnostic screening for PAH among CTDs patients.

## 1. Introduction

Pulmonary hypertension (PH) is a potentially life-threatening complication of connective tissue diseases (CTDs) in general, being particularly frequent in the clinical course of systemic sclerosis (SSc) [1]. Indeed, SSc is burdened by an excess of mortality, which is mainly attributable to cardiopulmonary involvement. The morbidity imposed by cardiovascular disease on these patients is also substantial and impacts on their quality of life [2]. The availability of effective treatments makes the early identification of PH of paramount clinical importance. In fact, life expectancy is significantly better among SSc patients with early rather than late detection of PH [3]. Therefore, there is a general consensus about the need for regular screening for PH in SSc patients [4]. The two-step DETECT algorithm is the most commonly used screening tool for PH, although novel biomarkers are needed to improve its diagnostic accuracy [5,6,7]. 

The mechanisms underlying PH development in CTDs are heterogeneous: they may be related to heart or lung involvement, to chronic thromboembolism, or finally, to a primary arteriolar vasculopathy (more appropriately defined as pulmonary arterial hypertension (PAH)) [8]. Vascular involvement is critical in the pathogenesis of SSc and it is driven by endothelial damage, impaired regulation of vessel tone, and proliferative vasculopathy [9]. Although the excessive fibroproliferative response to tissue injury characterizing SSc can become sustained independently from significant ongoing inflammation [10], the immune system has a pivotal role with B and T cells being central pathogenetic actors [11]. The activation of naive T cells requires a first signal involving the recognition by the T-cell receptor of a given antigen and a second non-antigen-specific co-stimulatory signal [12]. Recently, a dysregulation of the co-stimulatory signal has been claimed as a potential pathogenetic moment for SSc [13]. 

Other than the classical co-stimulatory pathway involving CD28 and its ligands CD80/CD86 [14], other pathways may play a co-stimulatory activity on T cells, among which is that involves the inducible co-stimulator (ICOS), highly expressed in activated T cells, and its ligand (ICOS-L), expressed by antigen-presenting cells (APC) [15]. Previous reports showed that ICOS serum levels and peripheral T-cell expression were increased in patients with early diffuse cutaneous SSc, and that the overexpression of ICOS leads to increased pro-inflammatory (IFN-γ, IL-17) and pro-fibrotic (IL-4) cytokines synthesis, fibroblast activation, and extracellular matrix synthesis [16,17].

In the present paper, we aimed at investigating whether the serum levels of ICOS and ICOS-L may act as potential biomarkers in the detection of PH in patients affected by CTDs.

## 2. Materials and Methods

We performed a cross-sectional, observational study on patients evaluated at the Pulmonary Hypertension Clinic of the Cardiology Division, University Hospital of Novara from 3 October 2016 to 12 December 2019. The study protocol was approved by the local ethical committee and conducted in strict accordance with the principles of the Declaration of Helsinki. Written informed consent was obtained from all individual participants included in the study.

We included patients older than 18 years, with a defined diagnosis of scleroderma-related disorders (SSc, mixed connective tissue disease—MCTD, scleroderma overlap syndromes, undifferentiated connective tissue disease—UCTD) according to the international diagnostic criteria. 

All the recruited patients underwent:-Clinical evaluation, including a comprehensive medical history and a physical examination performed by an experienced clinician;-A biochemistry panel including: complete blood count, creatinine and estimated glomerular filtration rate (eGFR), alanine aminotransferase and aspartate aminotransferase, gamma glutamyl transferase, uric acid, and brain natriuretic peptide (BNP);-A 12-lead electrocardiogram with 6-limb and 6 precordial leads with a paper speed set at the standard rate of 25 mm/s;-A transthoracic echocardiography (TTE) performed using the Vivid 7 or E9 cardiovascular ultrasound machine by GE Medical Systems (Horten, Norway) with a 1.7/3.4 MHz tissue harmonic transducer. All data were obtained in standardized patient positions, according to the standards of the American Society of Echocardiography. The test was performed by an expert cardiologist with special interest on pulmonary hypertension. The following parameters were generated: systolic pulmonary pressure (sPAP), right atrium area (RAA), right ventricle diameter (RVD), and ejection fraction (EF). Right ventricle systolic function was evaluated by estimating the tricuspid annular plane systolic excursion (TAPSE).

According to international guidelines, those patients with a suspected PAH underwent right heart catheterization. PAH was defined by mean pulmonary artery pressure (mPAP) ≥ 25 mmHg, pulmonary capillary wedge pressure ≤ 15 mmHg, and pulmonary vascular resistance > 3 wood units. Whenever contraindications to RHC occurred, pulmonary hypertension was diagnosed based on echocardiography-estimated sPAP ≥ 35 mmHg and additional high-probability criteria, in agreement with the 2015 ESC/ESR guidelines [18].

To evaluate whether ICOS and ICOS-L serum concentration were associated with CTD-related PAH or, more generally, to PAH, we also recruited a group of patients affected by idiopathic PAH. 

Blood samples were collected and centrifuged for 10 min at 4 °C at 3000 rpm. The samples were stored in the freezer at −80 °C. The ELISA quantitative assay of ICOS and ICOS-L in the human serum of patients included in this study were performed following the manufacturer’s instructions (Cloud-Clone Corp., Katy, TX, USA).

### Statistical Analysis

Anthropometric, clinical, and biochemical data were recorded in a database and analyzed using the statistical software package MedCalc v.19.6.4 (MedCalc Software, Broekstraat 52, 9030, Mariakerke, Belgium). The normality of ICOS and ICOS-L distribution was assessed by the Shapiro–Wilk test. Continuous variables are presented as medians and interquartile range (IQR). Differences in these variables between CTD, CTD-PAH, and PAH patients were compared by the Kruskal–Wallis test. The association between sPAP and the two biomarkers was assessed by Spearman’s rank correlation. Logistic regression models were run to evaluate the association of the selected biomarkers with PAH after correcting with different covariates, which might modify the serum concentration of ICOS and ICOS-L (age, gender, and renal function). To test the diagnostic performance of ICOS and ICOS-L in identifying patients with PAH, receiver operating characteristics curves were built with a calculation of the areas under the curve (AUC).

The level of significance chosen for all statistical analyses was 0.05 (two-tailed).

## 3. Results

We enrolled 109 patients (95 women, 87%); the median age in the study population was 63 (54–71) years. 

The study population was divided in three groups according to the underlying diagnosis: patients affected by connective tissue disease (CTD; N = 84, 77.0%), patients with CTD complicated with PAH (CTD-PAH; N = 13, 11.9%), and patients with isolated PAH (N = 12, 11.1%). Among those affected by CTD, the most common rheumatological diagnosis was SSc (N = 74, 76.3%); 6 patients were affected by MCTD (6.2%), 7 (7.2%) by UCTD; and finally, 10 (10.3%) by an overlap syndrome. The main parameters in the three study population groups are shown in Table 1. 

As expected, patients with PAH, independently from the underlying cause, are older; moreover, they were characterized by a significantly lower glomerular filtration rate. Looking at the echocardiographic parameters, sPAP and right atrium area were increased in the case of PAH. Interestingly, patients with both CTD-related and unrelated PAH showed higher ICOS and ICOS-L plasma concentrations. Conversely, the ratio was similar among groups. Furthermore, both ICOS (ρ = 0.351, CI 95% 0.165–0.513, *p* = 0.0004) and ICOS-L (ρ = 0.316, CI 95% 0.128–0.481, *p* = 0.001) were positively correlated to the sPAP.

We then tested the potential role of ICOS and ICOS-L in the identification of PAH among CTD patients. In Figure 1, we reported the difference in serum concentration of ICOS and ICOS-L between CTD and CTD-PAH patients, with the latter showing higher levels of both biomarkers.

To verify whether ICOS and ICOS-L were independently associated to the diagnosis of PAH in CTD patients, we run two logistic regression models (see Table 2).

As shown, ICOS and ICOS-L were independently associated with PAH. Finally, we built the corresponding ROC curves (Figure 2). Both ICOS (AUC: 0.843, CI 95% 0.754–0.909; *p* < 0.0001) and ICOS-L (AUC: 0.815, CI 95% 0.724–0.887; *p* < 0.001) demonstrated a good diagnostic performance in detecting PAH among CTD patients, similar to those of well-consolidated biomarkers of PH, such as BNP and uric acid (AUC: 0.881, CI 95% 0.796–0.939 and AUC: 0.711, CI 95% 0.606–0.801; *p* = n.s. for all comparisons). The best threshold for ICOS was 336 pg/mL; a value higher than this cut-off is 69.2% (CI 95% 38.6–90.9%) sensitive and 85.4% (CI 95% 75.8.6–92.2%) specific for the detection of PAH among CTD patients. Considering its use as a screening strategy, we also looked for the most conservative threshold; a value > 165 pg/mL is, in this setting, 100.0% (CI 95% 75.3–100.0%) sensitive and 50.0% (CI 95% 38.7–61.3%) specific.

The best threshold for ICOS-L was 4104 pg/mL; a value higher than this cut-off is 84.6% (CI 95% 54.6–98.1%) sensitive and 76.2% (CI 95% 65.7–84.8%) specific for the detection of PAH among CTD patients. 

## 4. Discussion

Pulmonary hypertension (PH) is a life-threatening complication of connective tissue disease (CTD) which worsens the disease prognosis but also contributes to a decline of patients’ quality of life. The availability of effective treatments makes the early diagnosis important to improve the prognosis by slowing down disease progression. Thus, the identification of novel tools for PH diagnosis is desirable: the present paper for the first time demonstrates a potential role for the soluble ICOS and ICOS-L in this specific context. 

Indeed, the main finding of our study is the observation that the soluble ICOS and ICOSL are both increased in patients with pulmonary arterial hypertension; it is particularly interesting that this finding is not dependent on the underlying cause of PAH. In fact, both CTD-PAH and isolated PAH patients showed increased levels of these biomarkers with respect to SSc patients. This is somehow surprising, since the ICOS-ICOSL pathway is related to immune activation and inflammation, and therefore expected to mark the autoimmune disease rather than the associated vasculopathy. However, Yanaba et al. already reported that patients affected by dSSc showed higher soluble ICOS concentrations than healthy subjects and that those with higher serum ICOS concentration more frequently showed lung involvement with more severe impairment of vital capacity [16]. 

These considerations, taken together, suggest a wider role for T cells in the development of lung vasculopathy. Recently, it has been found that Tregs are essential for regulating the course of PAH: Tregs can regulate PAH initiation and progression by secreting cytokines, interacting with other immune cells to ameliorate pulmonary arterial endothelial cells injury, regulating proliferation and apoptosis of smooth muscle cells, and fibroblasts [19]. It is therefore possible that ICOS+ Tregs may drive lung vasculopathy in SSc; Hasegawa et al. already reported that ICOS expression levels were specifically increased on both peripheral blood memory T cells and regulatory T cells (Tregs) from early dcSSc patients compared with those from healthy controls [17]. Consistently, ICOS+ Tregs are involved in different immune diseases, including systemic lupus erythematosus (SLE), rheumatoid arthritis (RA), sarcoidosis, with potential prognostic implications [20]: an elevated frequency of ICOS+ Tregs is directly correlated with SLE disease activity index scores and the serum antibody titer of anti-dsDNA [21]; moreover, a significantly larger proportion of ICOS+ Tregs was correlated to methotrexate unresponsiveness in RA patients [22]. Finally, a high expression level of ICOS was demonstrated in lung Tregs of pulmonary sarcoidosis patients [23]. 

T cells expressing ICOS may directly interact with endothelial cells expressing ICOS-L; notably, it has been shown that stressful stimuli, such as advanced glycation end products (AGEs), can lead to overexpression of ICOS-L on endothelial cells leading to endothelial dysfunction [24]. In synthesis, it may be postulated that the increased levels of soluble ICOS and ICOS-L may reflect a local overexpression in the lungs of ICOS non-Tregs and ICOS-L on endothelial cells, which may have a pathogenetic role in the development of PAH.

Aside from this putative model, the main implication of our findings is that both ICOS and ICOS-L may serve as diagnostic markers for PAH in general, and more specifically, for its detection among SSc patients. To the best of our knowledge, this is a novel finding. However, our study has limitations: the sample size is small and heterogeneous; thus, our observations need to be verified on larger cohorts. Moreover, the patients were already on treatment for the PAH and/or for the underlying CTD and further confounders might not have been taken into consideration; all these variables may have biased our observations. Thus, confirmation of this proof-of-concept study should come from other cohorts investigated as soon as the diagnosis is made. Furthermore, the number of RHCs performed in a close temporal window with measurement of ICOS and ICOS-L was very small and precluded us to evaluate the potential association between the two biomarkers and hemodynamic parameters directly measured by RHC. Finally, future studies should address the real impact of these novel biomarkers in the diagnostic process of PAH, particularly assessing whether their evaluation may really improve the diagnostic performance of the tools currently used.

In conclusion, the present study represents the basis for further investigation about the role of soluble ICOS and ICOS-L in the diagnosis of PAH in the context of CTDs. Moreover, it may represent a clue for the potential implication of this pathway in the pathogenesis of PAH.

## Figures and Tables

**Figure 1 diagnostics-12-00704-f001:**
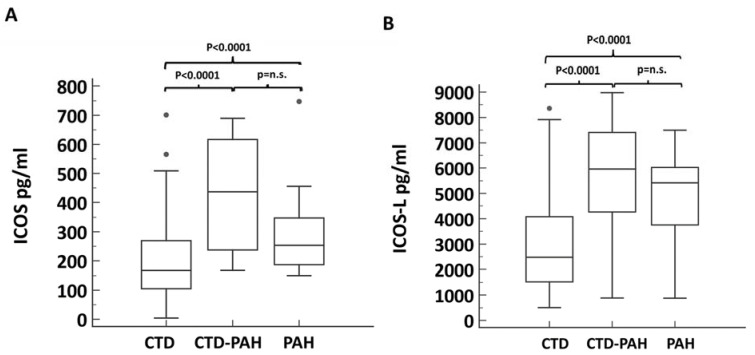
Comparison of ICOS and ICOS-L between patients affected by CTD, with and without PAH. In (**A**) the plasma concentration of ICOS is compared between groups; (**B**) conversely reports the comparison of ICOS-L plasma concentrations. Abbreviations: CTD = connective tissue disease and PAH = pulmonary arterial hypertension.

**Figure 2 diagnostics-12-00704-f002:**
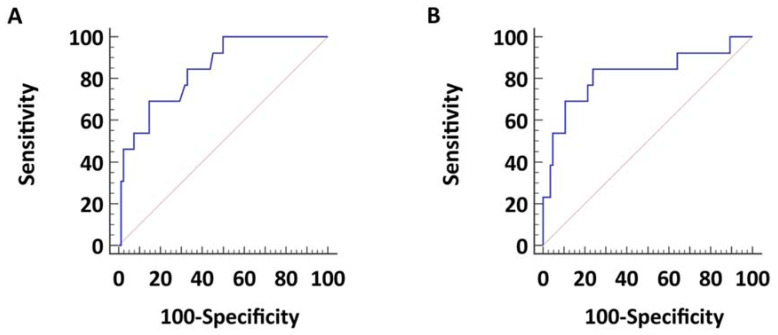
ROC curves for ICOS and ICOS-L. The figures represent the ROC curves for ICOS (**A**) and ICOS-L (**B**) for the identification of PH.

**Table 1 diagnostics-12-00704-t001:** General characteristics of study population groups. The table shows the main demographic features, frequencies, and percentage of main parameters in the 3 study population groups. Categorical variables are shown as frequencies (%), while continuous variables are shown as medians and interquartile range (IQR). Abbreviations: CTD: connective tissue disease; PAH: pulmonary arterial hypertension; Hb: hemoglobin; eGFR: estimated glomerular filtration rate; ALT: alanine aminotransferase; AST: aspartate aminotransferase; GGT: gamma glutamyl transferase; sPAP: systolic pulmonary artery pressure; RAA: atrium area; ICOS: inducible co-stimulator; ICOS-L: inducible co-stimulator ligand. * = CTD vs. CTD + PAH and PAH.

	CTD N = 84	CTD + PAH N = 13	PAH N = 12	*p*
Age (years)	62 (51.5–70)	69 (67–76)	73 (62–77)	0.004 *
Female	75 (68.8)	12 (11)	8 (73)	0.07
Hb (mg/dL)	13.2 (12.0–14.0)	12.0 (11.9–14.0)	11.0 (8.9–13.2)	0.13
eGFR	94 (76–102)	69 (50–78)	60.5 (48.5–79)	0.0001 *
ALT (IU/L)	18 (14–23)	16 (10–22)	14 (10–25)	0.35
AST (U/L)	22 (20–26)	21 (19–27)	22 (17–28)	0.72
GGT (U/L)	18 (13–33)	21 (15–89)	29 (21.5–46.5)	0.17
Uric acid (mg/dL)	4.7 (3.7–5.5)	6.2 (4.8–7)	6.3 (5.6–7.6)	0.002 *
BNP (pg/mL)	41 (24–80)	191 (127–527)	197 (43–372)	<0.0001 *
sPAP (mmHg)	26.5 (23.5–29.5)	50 (42–58)	56 (52.5–63.5)	0.0001 *
RAA (cm^2^)	13 (11–15)	20 (17–23)	25 (18–30)	0.0001 *
ICOS-L (pg/mL)	2450 (1500–4100)	6000 (4300–7000)	5004 (3750–6050)	0.0001 *
ICOS (pg/mL)	170 (105–275)	440 (240–600)	255 (190–345)	0.0001 *
ICOS-L/ICOS (pg/mL)	12.5 (7.4–24.6)	12.6 (10.3–17)	17 (15.4–25)	0.57

**Table 2 diagnostics-12-00704-t002:** Multivariate analysis of variables associated with PH. The table shows two multivariate analyses investigating the association of ICOS and ICOS-L with the diagnosis of PH. Abbreviations: eGFR: estimated glomerular filtration rate; ICOS: inducible co-stimulator; ICOS-L: inducible co-stimulator ligand.

**Variable**	** *p* **	**OR (95% CI)**
Age	0.68	1.0221 (0.9212–1.1341)
eGFR	0.46	0.9817 (0.9350–1.0307)
Gender	0.89	0.8249 (0.0581–11.7083)
ICOS-L	0.014	1.0005 (1.0001–1.0008)
**Variable**	** *p* **	**OR (95% CI)**
Age	0.63	1.0248 (0.9266–1.1333)
eGFR	0.32	0.9766 (0.9320–1.0235)
Gender	0.48	2.4763 (0.2032–30.1772)
ICOS	0.005	1.0061 (1.0018–1.0103)

## Data Availability

Data are available upon reasonable request to the corresponding author.

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
