# Peer review of "Increased Levels of ICOS and ICOSL Are Associated to Pulmonary Arterial Hypertension in Patients Affected by Connective Tissue Diseases"

_diagnostics, 2022, doi:10.3390/diagnostics12030704_

Round 1
Reviewer 1 Report
In this study, Bellan et al. analyzed serum levels of ICOS and ICOS-L in patients with connective tissue diseases with or without associated pulmonary hypertension as well as in pulmonary arterial hypertension patients. The authors performed a comprehensive clinical assessment of the 109 patients included. However, besides this clinical characterization, only 2 serum markers were determined to address the issue of potential novel biomarkers resulting in a low scientific merit of this study. It would have been useful to expand the mentioned "biochemistry panel" to further markers of interest. The manuscript is well written, owever there a numerous spelling mistakes requiring correction. I have some comments that should be addressed before publication.
1) Please specify "biochemistry panel" in line 88.
2) Did ICOS- and ICOS-L-levels correlate with the levels of hemodynamic parameters measured by RHC such as PVR, PAPm, SvO2. ,,, ?
3) In my eyes, Table 1 appears unnecessary. It should be deleted or integrated as a column to Table 2.
4) Are there any hints from the literature that ICOS-/ICOS-L-levels might correlate with renal insufficiency?
5) Can you provide details on the calculation of Odds ratios? The values mentioned in table 3 are difficult to understand for me. Usually, OR of 1 indicates that there is no difference between groups. Please specify in the methods and results sections.
6) Figure 2 needs correction with regard to the scales.
7) Please introduce the term "Tregs" in the discussion, line 202.
Author Response
First of all, I would like to thank the reviewer 1 for the valuable suggestions. Please find attached a point-to-point rebuttal:
Please specify "biochemistry panel" in line 88.
We have specified which biochemistry tests were included in the panel
Did ICOS- and ICOS-L-levels correlate with the levels of hemodynamic parameters measured by RHC such as PVR, PAPm, SvO2. ,,, ?
Unfortunately, the number of RHC performed in a short time window with ICOS and ICOS-L measurement was low and not adequate to obtain a reliable result. We have acknowledged this in the discussion section. For completeness, however, neither ICOS nor ICOS-L correlated with PVR, PAPm (ρ=0.036, p=0.90; ρ=0.342, p=0.24, respectively). The only potentially promising association was between ICOS-L and PVR (ρ=0.515, p=0.05); the correlation between PVR and ICOS was not significant (ρ=0.093, p=0.74) These calculations are based on N.=15 RHC.
In my eyes, Table 1 appears unnecessary. It should be deleted or integrated as a column to Table 2.
We have deleted Table 1 as suggested by the reviewers.
Are there any hints from the literature that ICOS-/ICOS-L-levels might correlate with renal insufficiency?
We did not find any specific effect of renal function on ICOS and ICOS-L levels; however, we considered appropriate to check for this variable when we built the logistic regression analysis.
Can you provide details on the calculation of Odds ratios? The values mentioned in table 3 are difficult to understand for me. Usually, OR of 1 indicates that there is no difference between groups. Please specify in the methods and results sections.
The issue was related to the approximation of the OR and it has been now corrected. As shown in the table, ICOS and ICOS-L showed 95%CI which did not include 1.00
Figure 2 needs correction with regard to the scales.
We have modified the Figure, as suggested.
Please introduce the term "Tregs" in the discussion, line 202.
We have defined the abbreviation as required.
Reviewer 2 Report
The authors performed quantitative measurement of serum inducible co-stimulator (ICOS) and its ligand (ICOS-L) in connective tissue diseases (CTD) patients with and without pulmonary arterial hypertension (PAH) and patients with isolated PAH in order to clarify the potential role of ICOS and ICOS-L as biomarkers of PH in CTD. They found that elevated both biomarkers were related to the diagnosis of PAH, suggesting their usefulness as diagnostic biomarkers for PAH among CTDs. Although these results are interesting, there are several important concerns as mentioned below for better understanding and improving of this study before the present study can be considered for publication.
- It seems quite important to compare the usefulness between novel and existing biomarkers. The authors showed serum uric acid levels but neither NT-proBNP nor BNP in Table 2. Non-inferiority of ICOS and ICOS-L compared with existing biomarkers should be evaluated. Were serum uric acid levels also useful as a diagnostic marker in the present study population? How about BNP or NT-proBNP? Moreover, additional significance of ICOS and ICOS-L for predicting PAH should be analyzed when they are evaluated in addition to BNP or NT-proBNP.
- The covariates in the Logistic regression analysis are another concern. The reason for selecting renal function as a covariate should be clearly described. In addition, blood uric acid, BNP, or NT-proBNP levels may be recommended as covariates.
- The authors mentioned that the CTDs in this study mainly comprise systemic sclerosis. It is necessary to clarify the proportion of each disease in the analyzed population. If the present study included no or few patients with systemic lupus erythematosus, mixed connective tissue disease, or Sjögren’s syndrome, the authors are recommended to use not CTDs but, for example, scleroderma spectrum diseases as a term of targeted disease.
Author Response
First of all we would likt to thank the reviewer for the valuable comments and suggestions; please find enclosed a point-to-point rebuttal:
It seems quite important to compare the usefulness between novel and existing biomarkers. The authors showed serum uric acid levels but neither NT-proBNP nor BNP in Table 2. Non-inferiority of ICOS and ICOS-L compared with existing biomarkers should be evaluated. Were serum uric acid levels also useful as a diagnostic marker in the present study population? How about BNP or NT-proBNP? Moreover, additional significance of ICOS and ICOS-L for predicting PAH should be analyzed when they are evaluated in addition to BNP or NT-proBNP
- We have added in table 1 and in table 2 the value of BNP, as suggested. BNP, as expected, as well as uric acid was significantly increased in patients affected by PH, independently from the underlying cause. We also compared the AUC of the correspondent ROC with the ones of ICOS and ICOS-L. The novel putative biomarkers showed similar diagnostic performance.
The covariates in the Logistic regression analysis are another concern. The reason for selecting renal function as a covariate should be clearly described. In addition, blood uric acid, BNP, or NT-proBNP levels may be recommended as covariates.
- We have added a comment in the methods section. We decided to include renal function, since there is no data available about the impact that renal insufficiency may have on ICOS and ICOS-L serum levels. Thus, we wanted to exclude the effect of renal function impairment, which is associated to PH. The sample size is too small to add all the suggested variables in a single model; we have tried to substitute renal function with uric acid and BNP. In the first model both ICOS (p=0.002) and ICOS-L (p=0.007), but not uric acid were independently associated to PH; according to the second model BNP was the only variable independently associated to PH, while ICOS (p=0.08) and ICOS-L (p=0.27) only approached statistical significance. However, with the logistic regression we mainly aimed to rule out the confounding effect of factors which notoriously may affect the association between a biomarker and a specific clinical condition, rather than evaluate the independent role of different biomarkers.
The authors mentioned that the CTDs in this study mainly comprise systemic sclerosis. It is necessary to clarify the proportion of each disease in the analyzed population. If the present study included no or few patients with systemic lupus erythematosus, mixed connective tissue disease, or Sjögren’s syndrome, the authors are recommended to use not CTDs but, for example, scleroderma spectrum diseases as a term of targeted disease.
- We have included the number of patients for any single rheumatological diagnosis in the results section.
Round 2
Reviewer 1 Report
The manuscript can be accepted in its current form.
Author Response
Thank tou very much for your comments
Reviewer 2 Report
This paper has been partly improved according to the reviewers' suggestions. However, sufficient additional analyses have not been performed yet. As the authors mentioned, it is important to rule out the confounding effect of specific clinical conditions. On the other hand, it is also essential to compare the diagnostic usefulness between novel and existing biomarkers when a novel biomarker is focused on as not a clue for understanding the disease pathophysiology but a diagnostic marker. Therefore, the 2 logistic regression models including BNP and uric acids, respectively, should be described in the revised manuscript.
- The additional analyses suggest that ICOS and ICOS-L are comparable with or tend to be inferior to BNP and tend to be superior to uric acids as a diagnostic marker for PAH although the results are not described in the revised manuscript. As a next step, the authors are recommended to analyze an impact of combination of BNP and ICOS/ICOS-L on the diagnosis of PAH. For example, when ‘BNP more than an appropriate cutoff value (50 or 300 pg/mL, or the value detected as the most appropriate in the ROC curve for BNP in the present population) and ICOS > 165 (or 336) pg/mL’ are set as one diagnostic marker, is the diagnostic accuracy for PAH improved compared with BNP alone? In the same way, whether ‘BNP more than an appropriate cutoff value or ICOS > 165 (or 336) pg/mL’ as one diagnostic marker improves the diagnosis of PAH or not is needed to be clarified.
Author Response
Thank you very much for acknowledging the improvement of the paper and for the additional comments.
We have modified the paper according to the reviewer's suggestions. More specifically:
- We have modified the logistic regression; we presented two models, in the first one, we included ICOS (or ICOS-L), age, gender, eGFR and uric acid. In the second one, we included ICOS (or ICOS-L), age, gender, eGFR and BNP. The results confirm that the novel biomarkers perform slightly worse than BNP and slightly better than uric acid. We added a comment in the discussion section.
- We also tried to assess the potential application of these biomarkers on top of others, already validated. We decided to use conservative cut-offs considering these biomarkers potentially promising in a screening strategy. IN this context, ICOS seems to be more promising to improve the specificity of BNP in a screening strategy. We added these results and a comment in the discussione section
Round 3
Reviewer 2 Report
This paper has been well revised according to the reviewers' suggestions, and is now suitable for publication in its present form.